# Double Check My Desired Return: Transformer with Value Validation for Offline RL

## Abstract

Recently, there has been increasing interest in applying Transformers to offline reinforcement learning (RL). Existing methods typically frame offline **R**L as a sequence modeling problem and learn actions **v**ia **S**upervised learning (RvS). However, RvS-trained Transformers struggle to align actual returns with desired target returns, especially when dealing with underrepresented returns in the dataset (interpolation) or missed higher returns that could be achieved by stitching suboptimal trajectories (extrapolation). In this work, we propose a novel method that **Do**uble **C**hecks the **T**ransformer with value validation for **O**ffline **R**L (*Doctor*). *Doctor* integrates the strengths of supervised learning (SL) and temporal difference (TD) learning by jointly optimizing the action prediction and value function. SL stabilizes the prediction of actions conditioned on target returns, while TD learning adds stitching capability to the Transformer. During inference, we introduce a double-check mechanism. We sample actions around desired target returns and validate them with value functions. This mechanism ensures better alignment between the predicted action and the desired target return and is beneficial for further online exploration and fine-tuning. We evaluate *Doctor* on the D4RL benchmark in both offline and offline-to-online settings, demonstrating that *Doctor* does much better in return alignment, either within the dataset or beyond the dataset. Furthermore, *Doctor* performs on par with or outperforms existing RvS-based and TD-based offline RL methods on the final performance.

## 1 Introduction

Transformer models (Vaswani, 2017) have dominated data-driven tasks such as natural language processing (Devlin, 2018; Brown, 2020; Achiam et al., 2023) and computer vision (Dosovitskiy, 2020; He et al., 2022) due to their ability to capture long-term dependencies and their effective scaling with data and compute (Kaplan et al., 2020; Rae et al., 2021). In recent years, there has been a growing interest in applying Transformers to reinforcement learning (RL) tasks, especially in the offline setting (Levine et al., 2020), where the agent learns from a fixed dataset of trajectories. To improve decision-making with Transformers, recent efforts have abstracted offline RL as a sequence modeling problem similar to large-scale language modeling, and have learned a policy via supervised learning. This approach is called reinforcement learning via supervised learning (RvS) (Emmons et al., 2021).

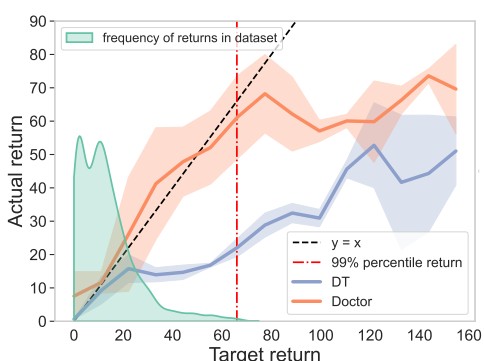

Figure 1: The achieved actual returns of *Doctor* and Decision Transformer (DT) conditioned on a wide range of target returns on the Hopper-Medium-Replay dataset. *Doctor* achieves much better alignment from the well-supported returns in the dataset to returns beyond the dataset.

RvS leverages the inherent stability and scalability of supervised learning to learn actions for each state based on the history trajectory, including the target returns. By specifying the policy's expertise

through the target return, the learned Transformer is expected to output actions that achieve the desired return (Chen et al., 2021; Janner et al., 2021). However, Transformers trained with RvS tend to struggle to align the actual return with the desired target return. As shown in Fig. 1, the Decision Transformer (DT) (Chen et al., 2021) learned by RvS can only achieve alignment within well-supported returns in the dataset. It fails to interpolate between underrepresented returns in the dataset (left side of the dashed red line) and to stitch information from multiple sub-optimal trajectories to achieve higher returns (right side of the dashed red line). This indicates that training a Transformer model with RvS is not sufficient to learn a policy with perfect alignment between the actual return and the target return. The use of supervised learning limits the policy's scope, and it lacks the capability to interpolate between underrepresented returns in the dataset, and to stitch information from multiple sub-optimal trajectories into a better one.

In this work, we propose a novel method that **Do**uble **C**hecks the **T**ransformer with value validation for **O**ffline **R**L (*Doctor*). *Doctor* integrates the strengths of supervised learning and TD learning by jointly predicting the actions and value functions. Supervised learning stabilizes the prediction of actions conditioned on target returns, while TD learning adds stitching capability to the Transformer and plays a critical role in aligning the actions with the target returns. At inference time, we introduce a double-check mechanism to first sample actions around target returns and then validate them with value functions. This mechanism ensures accurate alignment between the predicted action and the target return, enabling the extraction of policies with varying performance levels. Achieving this is valuable in scenarios like game AI, where NPCs with diverse skill levels are essential for balanced gameplay (Tanaka et al., 2024). Additionally, it enhances the model's ability to fine-tune its performance through ongoing online exploration. We evaluate our method on the D4RL benchmark (Fu et al., 2020) in both offline and offline-to-online settings, demonstrating that *Doctor* achieves much better return alignment, either within the dataset (interpolation) or beyond the dataset (extrapolation). Furthermore, *Doctor* performs on par with or outperforms existing RvS-based and TD-based offline RL methods on the final performance. Our contributions are summarized as follows:

- We propose a novel method, *Doctor*, that integrates the strengths of supervised learning and TD learning in a Transformer for offline RL. We jointly optimize the action prediction and value function to enhance the model's sequence modeling and stitching capabilities.

- *Doctor* introduces a double-check mechanism at inference time. We first sample actions around desired target returns and validate them with value functions to ensure accurate alignment. This mechanism allows the model to interpolate and extrapolate from the dataset and is beneficial for further online exploration and fine-tuning.

- We evaluate *Doctor* on the D4RL benchmark in both offline setting and online fine-tuning. We show that *Doctor* achieves much better return alignment either within the dataset or beyond the dataset, which is desired in return-conditioned models. Furthermore, *Doctor* also performs on par with or outperforms existing RvS-based and TD-based offline RL methods on the final performance.

## 2 PRELIMINARIES

### 2.1 REINFORCEMENT LEARNING

Reinforcement learning (RL) (Sutton & Barto, 2018) is a paradigm of agent learning via interaction. It can be modeled as a Markov Decision Process (MDP), a 5-tuple $\mathcal{M} = (\mathcal{S}, \mathcal{A}, \mathcal{R}, P, \gamma)$. $\mathcal{S}$ denotes the state space, $\mathcal{A}$ denotes the action space, $P(s'|s, a) : \mathcal{S} \times \mathcal{A} \times \mathcal{S} \to [0, 1]$ is the environment dynamics, $\mathcal{R}(s, a) : \mathcal{S} \times \mathcal{A} \to \mathbb{R}$ is the reward function which is bounded, $\gamma \in [0, 1]$ is the discount factor. Consider the finite horizon setting, the agent interacts with the environment for $T$ steps. Denote the state, action and reward at timestep $t$ as $s_t$, $a_t$ and $r_t$, a trajectory is a sequence of states, actions and rewards $\tau := (s_0, a_0, r_0, s_1, a_1, r_1, \ldots, s_T, a_T, r_T)$. The return at timestep $t$ is defined as $R_t = \sum_{i=t}^{T} \gamma^{i-t} r_i$. The goal of an RL agent is to learn an optimal policy $\pi$ that maximizes the expected return $R_0 = \mathbb{E}_\pi[\sum_{i=0}^{T} \gamma^i r_i]$.

In offline RL, instead of interacting with the environment, the agent learns from a static dataset of trajectories $\mathcal{D} := \{\tau_j\}$ such as the D4RL benchmark (Fu et al., 2020). The dataset is collected by an unknown behavior policy or policies, and the agent's goal is to learn a policy based on the dataset that

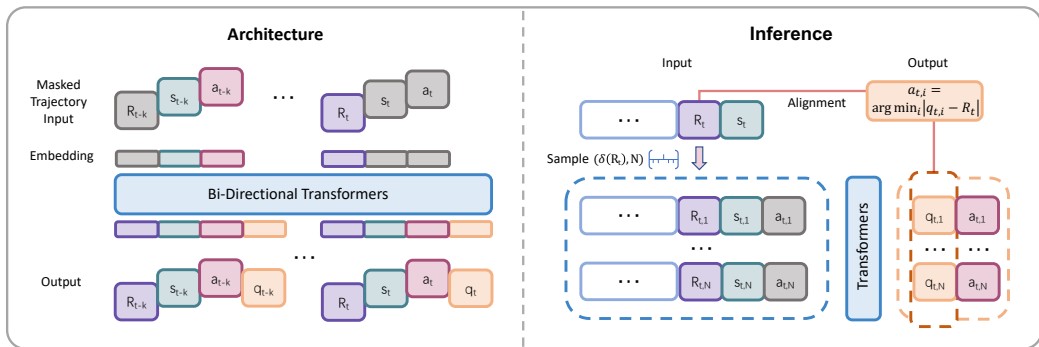

Figure 2: An overview of *Doctor*. (Left) The training involves reconstructing the original trajectory and estimating the action-value from a partial, randomly masked trajectory. Returns, states and actions are fed into modality-specific embeddings and then processed by the Transformers. The value heads estimate the action-value at each timestep. (Right) At inference time, *Doctor* samples actions around the target return and validates them with the value head outputs to ensure alignment.

performs well in the environment. This setting eliminates the need for online exploration which is practical in scenarios where exploration is expensive or dangerous, but it also introduces challenges as it removes the access to additional feedback from the environment (Levine et al., 2020).

## 2.2 TRANSFORMERS

The Transformer model (Vaswani, 2017) is a sequence-to-sequence model that uses self-attention mechanism to capture long-range dependencies in sequential data. The self-attention mechanism projects the input sequence into three vectors: query $Q$, key $K$ and value $V$, and computes the attention weights as

$$\text{Attention}(Q, K, V) = \text{softmax}(\frac{QK^T}{\sqrt{d_k}})V, \tag{1}$$

where $d_k$ is the dimension of the key vector. Transformers consist of multiple layers of multi-head self-attention and feed-forward neural networks. The model is trained with a masked language modeling objective, where the model predicts the next token in the sequence given the previous tokens. Transformers have shown remarkable success in various tasks such as natural language processing and computer vision.

Decision Transformer (DT) (Chen et al., 2021) applies Transformers to offline RL. Different from offline RL methods based on temporal difference learning, DT models the offline RL problem as a sequence modeling problem and learns a policy autoregressively by predicting the next action given the history trajectory conditioned on a target return. This set of approaches abstract offline RL as a sequence modeling problem and learns a policy via supervised learning (RvS) (Emmons et al., 2021). These approaches (Ghosh et al., 2021; Lee et al., 2022; Liu & Abbeel, 2023; Wu et al., 2023b) commonly condition on goals or target returns and expect the derived policy could be improved when feeding a high goal or target return. However, these methods struggle to align the actual return with the desired target return, especially when dealing with underrepresented returns in the dataset or missed higher returns that could be achieved by stitching sub-optimal trajectories.

## 3 METHOD

We outlined the limitations of RvS in the previous discussion. In this section, we introduce our method, *Doctor*, that integrates the strengths of supervised learning and TD learning in a Transformer for offline RL. *Doctor* aims to enhance the Transformer's stitching capabilities and improve the alignment between actual returns and desired target returns by leveraging value functions to double check predicted actions. We first introduce our model architecture in Section 3.1. Next, we describe how the model is jointly trained to predict actions and value functions in Section 3.2.

In Section 3.3, we introduce a double-check mechanism at inference time to improve alignment, which can also be used for online exploration and fine-tuning. Fig. 2 illustrates our method.

### 3.1 MODEL ARCHITECTURE

Following the common practice in prior work (Chen et al., 2021; Janner et al., 2021), we treat the offline RL as a sequence modeling problem. The trajectory $\tau$ consists of three modalities:

$$\tau = (R_0, s_0, a_0, R_1, s_1, a_1, \cdots, R_T, s_T, a_T), \tag{2}$$

where $R_t$ is the (discounted) return at time step $t$, $s_t$ is the state, and $a_t$ is the action.

Our model adopts an encoder-decoder architecture as a universal representation extractor. Both the encoder and decoder are bidirectional transformers, which are adept at capturing dependencies in sequential data. The task is based on sequence reconstruction from masked views (He et al., 2022), where a random subset of the sequence is masked and the model is tasked with reconstructing the original trajectory. This approach encourages the model to learn representations that capture the environment's dynamics and improves its ability to model the data. We apply random mask $M$ to certain elements of the sequence,

$$M(\tau) = (R_0, \_, a_0, R_1, s_1, \_, \cdots, \_, s_T, a_T), \tag{3}$$

where the masked elements are denoted as $\_$. Each type of element is embedded into a shared representation space using independent learnable linear embeddings. The masked sequence $M(\tau)$ is then fed into the encoder-decoder architecture $E$ and $D$ to obtain the last layer's latent representation $\tau^z = D(E(M(\tau)))$. A linear layer for each modality is applied to the latent representation $\tau^z$ to predict the return, state, and action at each timestep. The encoder-decoder processes the (masked) full sequence of latent representations and is trained to recover the original trajectory sequence $\tau$.

In addition to reconstructing the trajectory, the latent representation $\tau^z$ is also used to predict the action-value $q_t$. $\tau^z$ integrates the information from several timestep, which is beneficial for partial observability in RL tasks. And the action-value $q_t$ endows the model with the ability to evaluate the return and stitch sub-optimal trajectories for policy improvement. At inference time, we fed the unmasked full trajectory into the model to obtain the predicted actions and action-values.

### 3.2 TRAINING

Our training consists of two purposes: (1) reconstructing the original trajectory sequence from the masked input trajectory, which is a self-supervised learning task, and (2) learning the action-value $q_t$ to enable the model for stitching and to improve the alignment, which is TD learning. We jointly optimize the model to minimize the reconstruction error and the TD error.

**Self-Supervised Learning.** The self-supervised learning task reconstructs the original trajectory sequence from the randomly masked input trajectory. Denoting the learnable parameters of embeddings and the encoder-decoder as $\theta$, inducing conditional probabilities as $P_\theta$, the objective is to minimize the negative log-likelihood of the original trajectory sequence given the masked input:

$$\mathcal{L}_{\text{recon}}(\theta) = -\sum_{t=0}^{T} (\log P_\theta(R_t|M(\tau)) + \log P_\theta(s_t|M(\tau)) + \log P_\theta(a_t|M(\tau))). \tag{4}$$

Here, we take the summation over the whole trajectory sequence, but due to the complexity of self-attention (Vaswani, 2017; Kitaev et al., 2020), we sample minibatches of sequence length $K$ in practice for training efficiency.

**TD Learning.** Besides reconstructing the trajectory, the model is trained to predict the action-value $q_t$ at each timestep $t$. The Q-value function takes the latent trajectory representation $\tau^z$ as input and outputs the action-value estimates. This allows the Q-value function to share the rich representations learned by the reconstruction task. The goal of the Q-value function is to learn an optimal Q function within the dataset, which benefits the model's stitching capability and alignment with the target returns.

To avoid querying the learned Q-value function on out-of-sample actions, we utilize the asymmetric least squares loss function (Kostrikov et al., 2022). Denote the learnable parameters of the Q-value

function as $\phi$, the loss function is defined as:

$$\mathcal{L}_{\text{TD}}(\phi) = \sum_{t=0}^{T-1} L_2^{\nu} \left( r_t + \gamma Q_{\phi,t+1}(\tau^z, a_{t+1}) - Q_{\phi,t}(\tau^z, a_t) \right), \tag{5}$$

where $r_t$ is the reward, $\gamma$ is the discount factor, $Q_{\phi,t}$ and $Q_{\phi,t+1}$ are the Q-value functions at time step $t$ and $t+1$, respectively. $L_2^{\nu}(u) = |\nu - \mathbb{1}(u < 0)|u^2$ is the asymmetric least squares loss function. For $\nu = 0.5$, the loss function is equivalent to the standard mean squared error loss. For $\nu > 0.5$, the loss function is asymmetric, which down-weights the contributions of values smaller than zero (Newey & Powell, 1987; Kostrikov et al., 2022).

We initialize Q-value functions $Q_{\phi}$ and train them jointly with the Transformers. The overall objective is to minimize the sum of the reconstruction loss and the TD loss:

$$\mathcal{L}(\theta, \phi) = \mathcal{L}_{\text{recon}}(\theta) + \mathcal{L}_{\text{TD}}(\phi). \tag{6}$$

Unlike previous methods Yamagata et al. (2023); Wang et al. (2024) that learn the value functions beforehand and train the SL model separately afterward, our approach jointly trains the entire model. This joint training allows the model to learn a more accurate representation of the data and facilitates the integration of the two learning paradigms.

### 3.3 INFERENCE TIME ALIGNMENT

Due to the different purposes of supervised learning and TD learning, given a tuple $(R_t, s_t, a_t, q_t)$, the return $R_t$ represents the expected return in the dataset when taking action $a_t$ at state $s_t$, while the action-value $q_t$ reflects the expected *best* return after stitching. We should expect that $q_t \geq R_t$ and there could be a gap between them. This gap presents the difference between policy evaluation of the unknown behavior policy that collected the dataset and the best possible policy after policy improvement by stitching. This motivates us to introduce a double-check mechanism during inference to ensure alignment between the predicted action and the target return.

**Offline Evaluation.** Formally, given the current state $s_t$, denote $R_t$ as the desired target return at timestep $t$. We define $\delta(R_t) := \{R : |R - R_t| \leq \delta\}$ as the set of returns within a distance $\delta$ from $R_t$. We randomly sample $N$ returns,

$$\{R_{t,1}, R_{t,2}, \cdots, R_{t,N}\} \sim \text{Sample}(\delta(R_t), N), \tag{7}$$

and construct $N$ trajectories by replacing $R_t$ with $R_{t,i}$ in the original trajectory. We then input these $N$ trajectories into the model and obtain $N$ predicted actions $\{a_{t,1}, a_{t,2}, \cdots, a_{t,N}\}$ and action-values $\{q_{t,1}, q_{t,2}, \cdots, q_{t,N}\}$. To ensure alignment, we select the action with the nearest action-value to $R_t$ as the final action:

$$a_{t,i} = \arg\min_i |q_{t,i} - R_t|. \tag{8}$$

After taking action $a_{t,i}$ and obtaining the reward $r_t$, we updates the desired target return $R_{t+1}$ to $(R_t - r_t)/\gamma$ and repeat the process for the next timestep. This double-check mechanism first ensures that the actions are sampled based on the desired target returns and then validates them with the value functions to ensure alignment. This mechanism allows the model to interpolate/extrapolate between underrepresented or missing returns in the dataset.

To achieve high returns, we can set an aggressive target return that even exceeds the best possible return. The model will sample actions based on the desired target returns and validate them with value functions, selecting the action with the highest value. Conversely, to obtain a specific moderate return, we can set that return as the target. The model will then double-check the predicted action and select the action with the nearest value to the target return, helping avoid policy collapse.

**Online Fine-tuning.** Furthermore, this double-check mechanism can be utilized for online exploration and fine-tuning. During online exploration, we can sample actions based on the desired target returns, indicating the area we wish to explore. The value functions then evaluate these actions, providing prior knowledge about the expected returns. For example, we can take actions from the Boltzmann distribution based on the value functions,

$$\pi(a_t|s_t) = \frac{\exp(\beta q_t(a_{t,i}))}{\sum_i \exp(\beta q_t(a_{t,i}))}, \tag{9}$$

where $\beta$ is a temperature parameter that controls the sharpness of the distribution. This resulting in an effective exploration strategy that integrates prior knowledge from the value functions and the desired target returns. The method for *Doctor* is summarized in Algorithm 1.

---

**Algorithm 1** Double Checks the Transformer with value validation for Offline RL (***Doctor***)

---

1: Initialize sequence buffer $\mathcal{D}$, Transformer models with weights $\theta$, networks $Q$ with weights $\phi$
2: // Training Phase
3: **for** number of training steps $c = 0$ to $C$ **do**
4:     Sample a batch of length $K$ trajectories $(\ldots, R_t, s_t, a_t, r_t)$ from sequence buffer $\mathcal{D}$
5:     Update the sum of Self-SL loss and TD Learning loss $\mathcal{L}(\theta, \phi)$ via Eq. (6)
6: **end for**
7: // Inference Phase
8: **for** environment steps $t = 0$ to $T$ **do**
9:     Initialize the environment $s_0 \leftarrow Env$
10:     Randomly sample $N$ returns via Eq. (7), and construct $N$ trajectories
11:     Select the action with the nearest action-value to $R_t$ according to Eq. (8)
12:     **if** online fine-tuning **then** Select action from Boltzmann distribution via Eq. (9)
13:     **end if**
14:     Execute the action $a_t$ in the environment and observe the reward $r_t$ and next state $s_{t+1}$
15: **end for**

---

## 4 EXPERIMENTS

In this section, we present experimental results to evaluate the performance of *Doctor* on the D4RL benchmark (Fu et al., 2020). We first introduce the benchmark datasets and baselines in Section 4.1. We then demonstrate the superiority of *Doctor* in return alignment in Section 4.2. Finally, we evaluate the performance of *Doctor* in offline RL settings and online fine-tuning in Section 4.3.

### 4.1 SETUP

**Testbeds.** We evaluate *Doctor* on the D4RL benchmark (Fu et al., 2020), which consists of various environments and datasets leveraging the MuJoCo simulator (Todorov et al., 2012). For offline training, we focus on Gym locomotion V2 tasks with dense rewards, specifically Walker2D, Hopper, and HalfCheetah. Each task is configured with three levels of dataset difficulty: Medium-Replay, Medium, and Medium-Expert. The Medium dataset corresponds to policies performing at approximately one-third of expert-level performance, while the Medium-Replay dataset contains the replay buffer from an agent trained to medium-level performance. The Medium-Expert dataset combines trajectories generated by both medium and expert policies. We follow prior work to report the normalized scores, with a score of 100 representing expert-level performance (Fu et al., 2020).

**Baselines.** Our baselines are selected to cover a wide range of offline RL methods, we divide them into three categories:

- RvS-based methods. We consider RvS-R (Emmons et al., 2021), DT (Chen et al., 2021), MTM (Wu et al., 2023a) and ODT (Zheng et al., 2022). RvS-R uses an MLP with two fully connected layers to predict actions. The DT policy is trained using a GPT-based architecture with an autoregressive mask. MTM employs a BERT-like architecture with a combination of random and autoregressive masks. ODT is a DT-based variant trained with sequence-level entropy regularization for offline-to-online fine-tuning.

- TD learning-based methods. We compare *Doctor* with CQL (Kumar et al., 2020), an offline RL method that adopts pessimistic action estimation, and IQL (Kostrikov et al., 2022), an in-sample multi-step dynamic programming approach.

- Combination of RvS and TD learning. We include EDT (Wu et al., 2023b) and QDT (Yamagata et al., 2023), both aiming to improve RvS-trained Transformer-based policies through trajectory stitching.

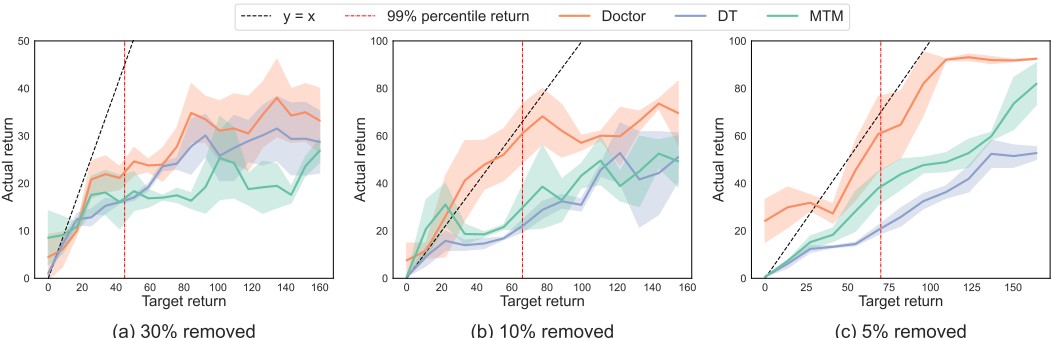

(a) 30% removed       (b) 10% removed       (c) 5% removed

Figure 3: We evaluate the alignment ability of *Doctor* on the hopper-medium-replay-v2 with the top X% returns of trajectories removed. The dashed red line presents the highest return in the dataset, and the dashed black lines denote the ideal alignment. *Doctor* achieves much better alignment across a wide range of target returns compared to DT and MTM.

## 4.2 THE SUPERIORITY OF *Doctor* IN RETURN ALIGNMENT

One of the key advantages of *Doctor* is its ability to achieve a wide range of desired target returns. This is the expected behavior for return-conditioned models that most existing methods fail to achieve. We evaluate the alignment ability of *Doctor* on the hopper-medium-replay-v2 dataset with varying levels of suboptimality. We remove trajectories with the top X% returns from the dataset and test the model across a wide range of target returns. As X% increases, the maximum returns of the trajectories in the dataset progressively decrease, moving the dataset further away from the optimal trajectory. We perform 250,000 gradient updates during training, and evaluate the model by rolling out 10 episodes. we set $N = 300$ and $\delta$ is small value fluctuating based on the maximum return in the dataset. Specifically, we choose $\delta = 5\% \times R_{\max}$. We report the final results over five random seeds.

We compare *Doctor* with DT and MTM, Fig. 3 shows the results. The x-axis represents the target return, the y-axis represents the actual return achieved in the environment. The dashed red line marks the 99% percentile return in the dataset. The dashed black lines denote the ideal line that perfectly aligns with the target return. Compared to DT and MTM, *Doctor* achieves much better alignment with the target return, even when the target return exceeds the maximum return to some extend in the dataset. This indicates that the integration of TD learning for target return alignment not only enables the transformer to interpolate more effectively within the dataset but also helps with extrapolation, achieving target returns more accurately even beyond those observed in the training data.

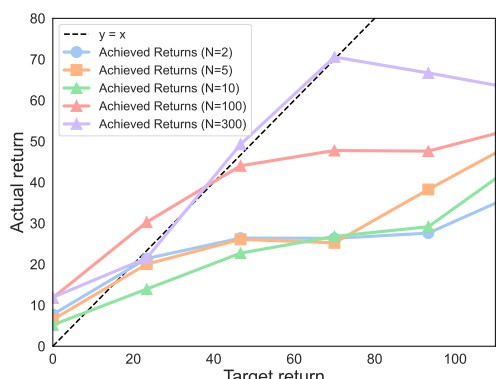

Figure 4: The effect of the number of samples $N$ in *Doctor*. As $N$ increases, *Doctor* achieves better alignment with the given target return.

We further analyze the impact of the number of samples $N$ on the performance of *Doctor* in Fig. 4. We test $N$ with increasing values from $\{2, 5, 10, 100, 300\}$, each generating a corresponding number of candidate actions. When $N = 2$, the model performs poorly, indicating that the target return alone is insufficient to ensure alignment. As $N$ increases, the model achieves better alignment, highlighting the importance of multiple samples for target return alignment. This demonstrates the effectiveness of the double-check mechanism in ensuring alignment between the predicted action and the target return.

| Environment | Dataset | RvS | CQL | IQL | DT | MTM | QDT | EDT | *Doctor* |
|---|---|---|---|---|---|---|---|---|---|
| HalfCheetah | Medium-Replay | 38.0 | **45.5** | 44.2 | 36.3 | 43.0 | 35.6 | 37.8 | 42.5 |
| Hopper | Medium-Replay | 73.5 | **95.0** | 94.7 | 82.7 | 92.9 | 52.1 | 89.0 | 93.2 |
| Walker2d | Medium-Replay | 60.6 | 77.2 | 73.9 | 66.6 | 77.3 | 58.2 | 74.8 | **79.9** |
| HalfCheetah | Medium | 41.6 | 44.0 | **47.4** | 42.0 | 43.6 | 42.3 | 42.5 | 43.5 |
| Hopper | Medium | 60.2 | 58.5 | 66.3 | 67.6 | 64.1 | 66.5 | 63.5 | **71.4** |
| Walker2d | Medium | 71.7 | 72.5 | **78.3** | 74.0 | 70.4 | 67.1 | 72.8 | 72.1 |
| HalfCheetah | Medium-Expert | 92.2 | 91.6 | 86.7 | 86.8 | 94.7 | - | - | **95.1** |
| Hopper | Medium-Expert | 101.7 | 105.4 | 91.5 | 107.6 | 112.4 | - | - | **112.7** |
| Walker2d | Medium-Expert | 106.0 | 108.8 | 109.6 | 108.1 | **110.2** | - | - | 109.3 |
| **Sum** | | 645.5 | 698.5 | 692.6 | 671.7 | 708.6 | - | - | **719.7** |

Table 1: Offline results on the D4RL benchmark. *Doctor* outperforms RvS-based methods like RvS, MTM, and DT in medium-level datasets such as Medium and Medium-Replay, and surpasses TD learning-based methods like IQL and CQL in expert-level datasets such as Medium-Expert. The highest scores among all methods are highlighted in bold.

### 4.3 OFFLINE AND ONLINE FINE-TUNING PERFORMANCE

To evaluate the final performance of *Doctor*, we compare it with baselines on the D4RL benchmark in both offline and online fine-tuning settings. We report the final results over five random seeds. Results for baseline methods are taken from the original papers. A detailed list of *Doctor*'s hyperparameters is summarized in Appendix D.2.

**Offline Results.** As shown in Table 1, *Doctor* achieves the strongest results in 4 out of 9 tasks and remains competitive in the remaining tasks. *Doctor* integrates both supervised learning and TD learning, allowing us to benefit from the strengths of both paradigms. In datasets with expert-level trajectories such as Medium-Expert, *Doctor* performs better than TD learning-based methods like IQL and CQL, demonstrating its ability of sequence modeling. In datasets with medium-level trajectories such as Medium and Medium-Replay, *Doctor* outperforms RvS-based methods like RvS, MTM, and DT, which indicates the stitching capability due to the value functions. This suggests that *Doctor* effectively integrates the advantages of both approaches, achieving superior performance with both low-return and high-return datasets. Due to space limit, we report the standard deviation of the results in Appendix D.2.

**Online Fine-tuning Results.** For online fine-tuning, we aim to test whether the model can further improve after interacting with the environment. We maintain the top 5% of the trajectories in the dataset and further interact with the environment for 200k steps, which corresponds to approximately 200 episodes. We sample actions based on Eq. (9) and set target returns as the maximum return $R_{\max}$ in the dataset and $\delta = 2R_{\max}, \beta = 100$. Each time after rolling out for one episode, we perform 200 gradient updates based on the collected data. The performance of ODT and IQL is taken from the ODT paper (Zheng et al., 2022)

As shown in Table 2, we observe a clear performance improvement when incorporating additional online interaction data. *Doctor* outperforms ODT and IQL on several tasks, with notable improve-

| Environment | Dataset | IQL | ODT | *Doctor* |
|---|---|---|---|---|
| HalfCheetah | Medium-Replay | **44.1 → 44.1** | 40.0 → 40.4 | 42.5 → 42.3 |
| Hopper | Medium-Replay | 92.1 → 96.2 | 86.6 → 88.9 | **93.2 → 97.1** |
| Walker2d | Medium-Replay | 73.7 → 70.5 | 68.9 → 76.9 | **79.9 → 81.4** |
| HalfCheetah | Medium | **47.4 → 47.4** | 42.7 → 42.2 | 43.5 → 43.8 |
| Hopper | Medium | 63.8 → 66.8 | **66.9 → 97.5** | 71.4 → 82.7 |
| Walker2d | Medium | 79.9 → 80.3 | 72.2 → 76.8 | **72.1 → 80.5** |
| **Sum** | | 401.0 → 405.3 | 377.3 → 422.7 | **402.6 → 427.8** |

Table 2: Online fine-tuning results. We report the average returns after 200k online interactions. *Doctor* observes notable improvements on Hopper and Walker.

ments in performance on Hopper and Walker. This demonstrates the double-check mechanism's effectiveness in online fine-tuning, allowing the model to explore more effectively.

## 5 RELATED WORK

Offline reinforcement learning (Levine et al., 2020) only uses existing data collected by unknown policies without additional online data collection. They aim to extract best possible policy from the existing dataset. One line of work is based on temporal difference (TD) learning (Zhang & Yu, 2020; Fujimoto et al., 2019; Wu et al., 2019; Kumar et al., 2019). To constrain the distance between the learned policy and the behavior policy to avoid distributional shift, they use a conservative value function to estimate the value of actions either by adding a regularization term in TD learning (Wu et al., 2019; Nair et al., 2020; Fujimoto & Gu, 2021; Wu et al., 2022), or updating the value function in an in-sample manner (Zhou et al., 2021; Kostrikov et al., 2022; Zhang et al., 2023; Xiao et al., 2023). CQL (Kumar et al., 2020) augments the standard Bellman error objective with a simple Q-value regularizer, such that the expected value of a policy under the learned Q-function lower-bounds its true value. Implicit $Q$-Learning (IQL) (Kostrikov et al., 2022) estimates the value of the best available action at a given state with expectile regression, without ever directly querying the $Q$ function for unseen actions. However, these methods are challenging to train and often require intricate hyperparameter tuning and various tricks to ensure stability and optimal performance across tasks (Sutton & Barto, 2018; Dong et al., 2020).

Another line of work is doing Reinforcement Learning via Supervised Learning (RvS) (Emmons et al., 2021). These method cast offline RL as a conditional sequence modeling problem and learn a policy autoregressively by predicting the next action by supervised learning (SL). Benefit from the inherent stability and scalability of SL, these methods bypasses the need for bootstrapping for long term credit assignment and avoids the "deadly triad" (Sutton & Barto, 2018) known to destabilize RL. These approaches (Janner et al., 2021; Ghosh et al., 2021; Lee et al., 2022; Liu & Abbeel, 2023; Wu et al., 2023b; Ma et al., 2024) commonly condition on goals or target returns and expect the derived policy derived could be improved when feeding a high goal or target return. DT (Chen et al., 2021) train a Transformer to autoregressively predict action sequences based on desired return and past trajectory. MTM (Wu et al., 2023a) applies masked prediction (Devlin, 2018; He et al., 2022) to learn a generic and versatile model for prediction, representation, and control. RADT (Tanaka et al., 2024) achieves precise alignment between actual returns and target returns by separating return and state-action sequences and introducing specialized aligners, which overcomes the attention allocation limitations of prior methods like DT. Although RvS tends to be stable, and scales well with compute and data, it fails to achieve one of the desired properties of offline RL agents, stitching. This property is an ability to combine parts of sub-optimal trajectories and produce an optimal one.

To enhance Transformers with stitching ability in offline RL, QDT (Yamagata et al., 2023) utilises the Dynamic Programming results to relabel the return-to-go in the training data to then train the DT (Chen et al., 2021) with the relabelled data. EDT (Wu et al., 2023b) optimizes the trajectory by retaining a longer history when the previous trajectory is optimal and a shorter one when it is sub-optimal, enabling it to stitch with a more optimal trajectory. QT (Hu et al., 2024) combines the trajectory modeling capabilities of Transformers with the predictive strengths of dynamic programming (DP) methods. Results on D4RL benchmarks show QT achieves state-of-the-art performance in offline RL. Our work aims to learn an accurate return-conditioned model, which require the model to extrapolate the return in the low-data regime, and also to stitch the trajectory to produce a more optimal one in the absence of the optimal trajectory.

## 6 CONCLUSION

In this work, we propose a novel method that Double Checks the Transformer with value validation for Offline RL (*Doctor*). *Doctor* integrates supervised learning and TD learning based on a Transformer architecture. Our method leverages the strengths of Transformers for sequence modeling and the joint optimization of Q-value functions enhances the model's ability of return alignment and stitching. At inference time, we introduce a double-check mechanism to sample actions based on desired target returns and validate them with value functions to ensure alignment. The double-check mechanism allows the model to interpolate and extrapolate from the dataset, and is also suitable for

online exploration and fine-tuning. Experiments on the D4RL benchmark demonstrate that *Doctor* achieves state-of-the-art performance compared to existing RvS-based and TD-based offline RL methods. We show that *Doctor* can effectively interpolate between underrepresented returns in the dataset and stitch information from multiple sub-optimal trajectories and produce a better one. Furthermore, *Doctor* demonstrates superior final performance both in offline RL settings and online fine-tuning, highlighting its effectiveness in a wide range of tasks.

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

# A ADDITIONAL JUSTIFICATION OF VALUE VALIDATION

Assume our action-value function is optimal, $Q = Q^*$. We select the action $a_t$ that minimizes the absolute difference between the predicted action-value and the desired return:

$$a_t = \arg\min_a |Q^*(s_t, a) - R_t|$$

We show that this action selection aligns the expected return with $R_t$, achieving return alignment.

**Case 1**: $R_t > R$ (desired return exceeds achievable return)

The desired return $R_t$ is greater than the maximum possible return $R = V^*(s_t)$, where:

$$V^*(s_t) = \max_a Q^*(s_t, a)$$

Since $Q^*(s_t, a) \leq V^*(s_t)$ for all actions $a$, we have:

$$Q^*(s_t, a) - R_t \leq V^*(s_t) - R_t < 0$$

The absolute difference $|Q^*(s_t, a) - R_t|$ is minimized when $Q^*(s_t, a)$ is maximized. The optimal action $a^*$ is:

$$a^* = \arg\max_a Q^*(s_t, a)$$

Selecting $a_t = a^*$ minimizes $|Q^*(s_t, a) - R_t|$. Even when $R_t$ is unattainable, the method selects the action that yields the highest possible return.

**Case 2**: $R_t < R$ (desired return less than achievable return)

The desired return $R_t$ is less than the maximum possible return $R = V^*(s_t)$. There may exist actions $a$ such that $Q^*(s_t, a) \approx R_t$. By minimizing $|Q^*(s_t, a) - R_t|$, we may select an action $a_t$ where $Q^*(s_t, a_t) \geq R_t$ but potentially less than $V^*(s_t)$. This action aligns the expected return with $R_t$ without necessarily maximizing it and allows for controlled performance, achieving the desired return.

# B ADDITIONAL ENVIRONMENT DETAILS

**D4RL Gym Locomotion**. The D4RL Gym locomotion benchmark (Fu et al., 2020) includes environments provided by OpenAI Gym (Brockman, 2016), specifically Walker2d, Hopper, and HalfCheetah. These environments are widely used for evaluating reinforcement learning algorithms. For expample, Walker2d environment simulates a robot tasked with walking as fast and as stably as possible. The robot must coordinate its two legs to achieve efficient locomotion without falling over. These environments are designed to test an agent's ability to learn complex motor skills and optimize control strategies.

**Adroit** (Rajeswaran et al., 2017). Adroit is a suite of dexterous manipulation tasks designed to simulate the control of a five-fingered robotic hand. Our experiments focus on three tasks from this suite: Pen, Door, and Hammer. For example, the Pen task involves rotating a pen to a specific orientation using the robotic hand's dexterous manipulation skills. The 'cloned' tasks used in our experiment collect a 50-50 split of demonstration data and 2500 trajectories sampled from a behavior cloning policy.

**Maze2D**. Maze2D is a navigation task where the agent is required to reach a fixed target position. These tasks are designed to evaluate the ability of offline reinforcement learning algorithms to 'stitch' together different trajectory fragments (Fu et al., 2020). We use three environments included in Maze2D, umaze, medium, and large, with complexity and path length to the target increasing sequentially.

# C ADDITIONAL EXPERIMENTAL RESULTS

## C.1 OFFLINE RESULTS

| Gym Tasks | RvS | CQL | IQL | DT | MTM | QDT | EDT | RADT | QT | Doctor |
|---|---|---|---|---|---|---|---|---|---|---|
| HalfCheetah-MR | 38.0 | **45.5** | 44.2 | 36.3 | 43.0 | 35.6 | 37.8 | 41.3 | 44.7 | 42.5±0.3 |
| Hopper-MR | 73.5 | 95.0 | 94.7 | 82.7 | 92.9 | 52.1 | 89.0 | **95.7** | 95.3 | 93.2±3.6 |
| Walker2d-MR | 60.6 | 77.2 | 73.9 | 66.6 | 77.3 | 58.2 | 74.8 | 75.9 | **91.5** | 79.9±2.2 |
| HalfCheetah-M | 41.6 | 44.0 | **47.4** | 42.0 | 43.6 | 42.3 | 42.5 | 42.8 | 45.3 | 43.5±0.8 |
| Hopper-M | 60.2 | 58.5 | 66.3 | 67.6 | 64.1 | 66.5 | 63.5 | **90.0** | 85.8 | 71.4±8.1 |
| Walker2d-M | 71.7 | 72.5 | 78.3 | 74.0 | 70.4 | 67.1 | 72.8 | 75.6 | **84.6** | 72.1±7.2 |
| HalfCheetah-ME | 92.2 | 91.6 | 86.7 | 86.8 | 94.7 | - | - | 93.1 | 92.0 | **95.1**±0.3 |
| Hopper-ME | 101.7 | 105.4 | 91.5 | 107.6 | 112.4 | - | - | 110.4 | 111.5 | **112.7**±0.4 |
| Walker2d-ME | 106.0 | 108.8 | 109.6 | 108.1 | **110.2** | - | - | 109.7 | 109.9 | 109.3±1.5 |
| **Sum** | 645.5 | 698.5 | 692.6 | 671.7 | 708.6 | - | - | 734.5 | **760.6** | 719.7 |

| Maze2D Tasks | BC | CQL | IQL | DT | MTM | BCQ | BEAR | TD3+BC | QDT | Doctor |
|---|---|---|---|---|---|---|---|---|---|---|
| maze2d-umaze-v1 | 88.9 | 94.7 | 42.1 | 31.0 | 58.0 | 49.1 | 65.7 | 14.8 | 57.3 | **117.2**±12.3 |
| maze2d-medium-v1 | 38.3 | 41.8 | 34.9 | 8.2 | 52.9 | 17.1 | 25.0 | 62.1 | 13.3 | **84.3**±7.6 |
| maze2d-large-v1 | 1.5 | 49.6 | 61.7 | 2.3 | 24.2 | 30.8 | 81.0 | **88.6** | 31.0 | 47.5±3.1 |
| **Sum** | 128.7 | 186.1 | 138.7 | 41.5 | 135.1 | 97.0 | 171.7 | 165.5 | 101.6 | **249.0** |

| Adroit Tasks | BC | CQL | IQL | DT | MTM | BCQ | BEAR | GDT | TD3+BC | Doctor |
|---|---|---|---|---|---|---|---|---|---|---|
| pen-cloned-v1 | 37.0 | 39.2 | 37.3 | 75.8 | 80.5 | 50.9 | 26.5 | 86.2 | 5.1 | **98.4**±15.5 |
| hammer-cloned-v1 | 0.6 | 2.1 | 2.1 | 3.0 | 5.3 | 0.4 | 0.3 | 8.9 | -0.3 | 5.9±3.1 |
| door-cloned-v1 | 0.0 | 0.4 | 1.6 | 16.3 | 17.4 | 0.0 | -0.1 | 19.8 | 0.2 | **24.2**±8.7 |
| **Sum** | 37.6 | 41.7 | 41.0 | 95.1 | 103.2 | 51.3 | 26.7 | 114.9 | 5.0 | **128.5** |

Table 3: Results on the D4RL benchmarks, Maze2D tasks, and Adroit tasks. Standard deviations are shown in smaller font to improve readability.

In the offline experiments, we introduce Maze2D and Adroit as additional test environments, as shown in Table 3. For the Gym tasks, we added RADT (Tanaka et al., 2024) and QT (Hu et al., 2024) as baselines. RADT addresses target return alignment. QT is the state-of-the-art method, which enhances stitching capability by incorporating the TD method. We implemented QT using their official code. For Maze2D and Adroit, we also added BCQ (Fujimoto et al., 2019), BEAR (Kumar et al., 2019), TD3+BC (Fujimoto & Gu, 2021), and GDT (Hu et al., 2023) as comparison baselines.

Notably, Doctor significantly outperforms other baselines on Maze2D, demonstrating its stitching capability. The Maze2D dataset consists of suboptimal trajectories and is specifically designed to evaluate stitching ability. It can be observed that supervised learning based method like DT, struggles when faced with suboptimal trajectories. Doctor improves performance by leveraging tempral difference learing. Furthermore, on the complex control tasks in the Adroit cloned environment, Doctor also achieves clear improvements compaired with other baselines.

## C.2 OFFLINE TO ONLINE RESULTS

In the offline-to-online experiments, we evaluate ODT and Doctor with 20k and 100k interaction data. The results are reported in Table 4 Doctor shows greater performance improvements with 100k interaction data compared to 20k. We also test Doctor on Adroit tasks, where offline-to-online methods are often used for evaluation on these tasks. Clear improvements were observed in online fine-tuning experiments on adroit. We include Cal-QL (Nakamoto et al., 2024), AWAC (Nair et al., 2020), and SPOT Wu et al. (2022) as baselines for comparison. The number for these baselines on adroit is reported from CORL Tarasov et al. (2024).

## C.3 ADDITIONAL VISUALIZATION FOR VALUE ALIGNMENT

We additionally visualize the effect of value alignment as shown in Fig. 5, showing 1x to 3x the maximum returns in the dataset, following the approach of DT (Chen et al., 2021).

| Gym Tasks | IQL (0.2m) | ODT (0.2m) | Doctor (0.2m) | AWAC (1m) | ODT (1m) | Doctor (1m) |
|---|---|---|---|---|---|---|
| HalfCheetah-MR | **44.1 → 44.1** | 40.0 → 40.4 | 42.5 → 42.3±2.3 | – | 40.0 → 41.2 | 42.5 → 42.7±1.6 |
| Hopper-MR | 92.1 → 96.2 | 86.6 → 88.9 | **93.2 → 97.1**±2.4 | – | 86.6 → 91.3 | **93.2 → 97.9**±3.3 |
| Walker2d-MR | 73.7 → 70.5 | 68.9 → 76.9 | **79.9 → 81.4**±4.7 | – | 68.9 → 78.4 | **79.9 → 84.2**±5.6 |
| HalfCheetah-M | **47.4 → 47.4** | 42.7 → 42.2 | 43.5 → 43.8±1.7 | 37.4 → 41.4 | 42.7 → 43.6 | 43.5 → 42.5±2.1 |
| Hopper-M | 63.8 → 66.8 | **66.9 → 97.5** | 71.4 → 82.7±10.5 | 72.0 → 91.0 | **66.9 → 98.1** | 71.4 → 88.5±9.1 |
| Walker2d-M | 79.9 → 80.3 | 72.2 → 76.8 | **72.1 → 80.5**±3.2 | 30.1 → 79.1 | 72.2 → 77.0 | **72.1 → 80.7**±5.9 |
| **Sum** | 401.0 → 405.3 | 377.3 → 422.7 | **402.6 → 427.8** | – | 377.3 → 429.6 | **402.6 → 436.5** |

| Adroit Tasks | AWAC | CQL | IQL | SPOT | Cal-QL | Doctor |
|---|---|---|---|---|---|---|
| pen-cloned-v1 | 88.7 → 86.8 | -2.8 → -1.3 | 84.2 → 102.0 | 6.2 → 43.6 | -2.7 → -2.7 | **98.4→110.5**±18.5 |
| door-cloned-v1 | 0.9 → 0.0 | -0.3 → -0.3 | 1.2 → 20.3 | -0.2 → 0.0 | -0.3 → -0.3 | **24.2→24.7**±9.9 |
| hammer-cloned-v1 | 1.8 → 0.2 | 0.6 → 2.9 | 1.4 → 57.3 | 4.0 → 3.7 | 0.3 → 0.2 | 5.9 → 46.8±28.9 |
| **Sum** | 92.4 → 87.0 | -2.8 → 1.0 | 86.8 → 179.9 | 9.8 → 47.1 | -3.0 → -3.1 | **128.5→182** |

Table 4: Results of Gym and Adroit tasks for online fine-tuning. The arrow indicates the change in performance.

Return alignment cannot be solved merely by using high target returns. Figures Fig. 5 demonstrate this clearly: when using target returns beyond the maximum return in the dataset (e.g., 1.0x to 3.0x maximum return), we observe that supervised learning based model's performance gradually saturates, failing to achieve the same level of alignment as Doctor.

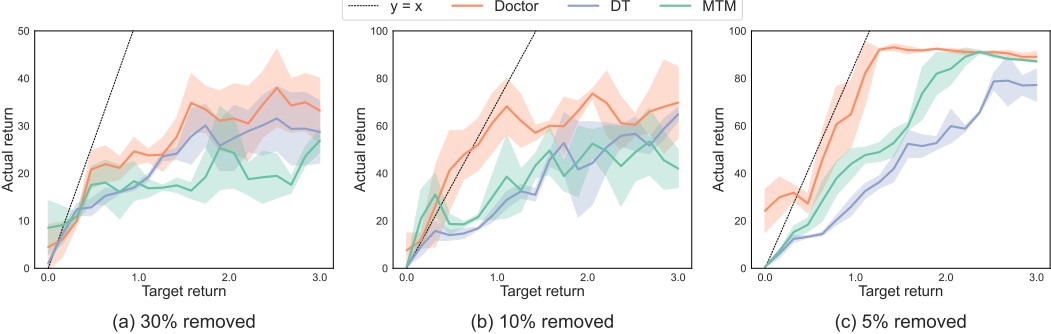

(a) 30% removed     (b) 10% removed     (c) 5% removed

Figure 5: We visualize the effect of value alignment, showing 1x to 3x the maximum returns in the dataset. *Doctor* achieves much better alignment across a range of target returns compared to DT and MTM.

# D  MODEL AND TRAINING DETAILS

## D.1  TRAINING DETAILS

We provide implementation details regarding *Doctor*. The Transformers include a bidirectional transformer encoder and a bidirectional transformer decoder. Before inputting the sequence data into the model, each input modality is projected into the embedding space through independent embed-encodings. The output of the decoder is connected to a 2-layer MLP with layer normalization, which is used to reconstruct the trajectory sequence. The Transformer is trained with a randomly sampled series of mask ratios similar to (Wu et al., 2023a): mask_ratios = [0.6, 0.7, 0.8, 0.85, 0.9, 0.95, 1.0]. For data sampling, we adopt a two-step sampling method similar to that used in DT (Chen et al., 2021), where we first sample a single trajectory and then uniformly sample sub-trajectories of a certain sequence length.

For offline training, we initialize the AdamW optimizer for the Transformer model and the Adam optimizer for the Q-value head, employing both warmup and decay schedules. The Q-value head consists of a single 512-dimensional MLP layer, which connects to the output of the Transformer decoder. All hyperparameters are summarized in Table 5.

During the fine-tuning stage, we initialize the replay buffer with the top 5% highest-return trajectories from the offline dataset. Each time we interact with the environment, we fully roll out one

episode using the current policy and add it to the replay buffer. We then update the policy and roll out again, following a process similar to (Zheng et al., 2022).

## D.2 MODEL HYPERPARAMETERS

Table 5: Hyperparameters

| Bidirectional Transformer | Value |
|---|---|
| Encoder layers | 2 |
| Decoder layers | 1 |
| Activation function | GeLu |
| Number of attention heads | 4 |
| Embedding dimension | 512 |
| layers of decoding head | 2 |
| Dropout | 0.10 |
| Positional encoding | Yes |
| Dropout | 0.1 |
| Learning rate | 0.0001 |
| Weight decay | 0.005 |
| betas | [0.9,0.999] |
| Learning rate warmup steps | 40000 |
| **Value function head $Q$** | **Value** |
| Number of layers | 1 |
| Activation function | ReLu |
| Embedding dimension | 512 |
| tau | 0.7 for gym tasks, 0.9 for maze2d and 0.8 for adroit |
| Learning rate | 0.0001 |
| Weight decay | 5e-4 |
| **General** | **Value** |
| Eval episodes | 10 |
| Input trajectory length | 4 for Gym tasks, 10 for maze2d and 12 for adroit |
| Trainning steps | 140000 |
| Batch size | 1024 |
| Discount factor | 0.99 |

## D.3 INFERENCE AND COMPUTATION OVERHEAD

Table 6: Comparison of Time Complexity for Different Algorithms

| Time Complexity | DT | MTM | QT | Doctor |
|---|---|---|---|---|
| **Inference** (seconds) | 0.01 | 0.056 | 0.016 | 0.065 |
| **Training** (seconds) | 2.13 | 1.27 | 2.51 | 1.34 |

We evaluate the inference and training time of different algorithms as follows. For training, we use a batch of size 2048 as input to all four algorithms and measure the time required to train one batch. For inference, we calculate the Frames Per Second (FPS) by running 1000 environment interaction steps, measuring the total time taken, and then dividing by 1000. The reported values are the averages of several test runs as shown in Table 6.

It can be observed that Doctor's computation overhead does not significantly increase. During inference, Doctor processes one batch at a time and leverages a unified architecture to generate both

action predictions and value estimates simultaneously, allowing for fully parallel computation. Doctor leverages a Q-function trained with the transformer-based representation, and compared to the computational cost of the transformer itself, the additional cost of Q-function computation is relatively low

