# OpenReview forum: "Double Check My Desired Return: Transformer with Value Validation for Offline RL"
_ICLR.cc/2025/Conference — Submitted to ICLR 2025_

### Official Review · Reviewer_PDqo · 2024-10-19

**Soundness:** 2
**Presentation:** 3
**Contribution:** 2
**Rating:** 6
**Confidence:** 3

**Summary:**

RvS-trained Transformers lacks 1) the capability to interpolate between underrepresented returns in the dataset, and 2) to stitch information from multiple sub-optimal trajectories into a better one, due to its supervised nature.
The paper introduces Doctor, which aligns the return of the policy with the target return, by selecting the action that has the nearest Q function (learned via expectile regression, similar to IQL) to the target returns.
Doctor shows much better return alignment across various target returns, and pars with / outperforms other RvS methods in terms of performance.

**Strengths:**

1. Doctor shows much better return alignment compared to previous RvS methods.

* Experimental results in Figure 1, Figure 3 shows that Doctor shows much better alignment.
* This alignment can extrapolate for some datasets (i.e. being better than the dataset).

2. The algorithm of Doctor is simple and intuitive.

* The alignment happens in the inference time, and the algorithm is straight-forward (choosing the action with the closest returns)
* Combines SL and IQL-like loss, which does not need to query out-of-distribution actions.

**Weaknesses:**

1. The experimental results is insuffcient on showing the stitching capability of Doctor.
* While stitching is one of the main reason of integrating TD-learning to SL, the performance improvement compared to MTM (which is a pure RvS method) is incremental (708.6 vs 719.7), according to Table 1.
* Generally, D4RL MuJoCo Gym environments is not a good benchmark to evaluate stitching capabilities. Experimental results in D4RL Maze2D tasks or AntMaze tasks, which are designed for stitching, will be helpful on evaluating the stitching capabilities of Doctor.

2. The experimental results is insufficient on showing that "better alignment in returns help offline-to-online fine-tuning".
* Experiment results in Table 2 shows that the performance increase during online fine-tuning is larger for ODT. It looks counterintuitive, considering superior alignment capability of Doctor.
* Again, D4RL MuJoCo Gym environments might be not a good choice to evaluate offline-to-online fine-tuning. For example, Cal-QL[1] uses AntMaze, Kitchen, Adroit tasks for their experiments; experimental results on these environments might better highlight the superiority of Doctor.

[1] Mitsuhiko Nakamoto et al., "Cal-QL: Calibrated Offline RL Pre-Training for Efficient Online Fine-Tuning", NeurIPS 2023.

**Questions:**

Q1. Does having better return alignment helps the final performance?
* i.e., can the problem of return alignment be solved by just having high target returns? (e.g. DT[2] uses 5x maximum returns in the dataset)
* It will be insightful if there are experimental results for ablating the inference-time alignment (with properly tuned target returns, e.g. 1x or 5x maximum return of dataset).

Q2. Does having better return alignment helps online fine-tuning?
* It will be exciting if Doctor is able to show clear improvements in online fine-tuning experiments on AntMaze or Kitchen or Adroit tasks, where offline-to-online methods (e.g. Cal-QL) have showed significant improvements.

Q3. Other than Q1 (final performance) and Q2 (online fine-tuning), can you share your ideas on why having better return alignment is beneficial, if there is any?
* e.g., does it help reducing the effort of searching for the target return used in inference?

Q4. Does Doctor improves stitching capability?
* Experimental results in D4RL Maze2D tasks or AntMaze tasks, which are specifically designed for stitching, would be informative to assess the stitching capability of Doctor.

Q5. What target return is used for evaluation?
* Can you share how did you find it? (e.g. is it the maximum return from the dataset?)


[2] Lili Chen et al., Decision Transformer: Reinforcement Learning via Sequence Modeling, NeurIPS 2021.

---

> ### Author Response · Authors · 2024-11-21
> **Part 1**
>
> **Weaknesses:**
> > 1. The experimental results is insuffcient on showing the stitching capability of Doctor.
> > While stitching is one of the main reason of integrating TD-learning to SL, the performance improvement compared to MTM (which is a pure RvS method) is incremental (708.6 vs 719.7), according to Table 1.
> > Generally, D4RL MuJoCo Gym environments is not a good benchmark to evaluate stitching capabilities.
>
> re:
>
> Thank you for your advice. We have conducted further experiments on some extra tasks in Table 3 (see appendix C.1 [703-729] in our revised version). Notably, in Maze2D, Doctor outperforms some other baselines by a clear margin, which emphasize stitching capabilities of Doctor.
>
> > 2. The experimental results is insufficient on showing that "better alignment in returns help offline-to-online fine-tuning".
> > Experiment results in Table 2 shows that the performance increase during online fine-tuning is larger for ODT. It looks counterintuitive, considering superior alignment capability of Doctor.
> > Again, D4RL MuJoCo Gym environments might be not a good choice to evaluate offline-to-online fine-tuning.
>
> re:
>
> We have conducted additional experiments on Adroit tasks regarding the evaluation of offline-to-online in Table 4 (see appendix C.2 756-770). The results show that Doctor achieves clear improvements on the Adroit tasks, demonstrating its fine-tuning capability in more challenging environments.
>
>
> **Questions:**
>
> > Q1. Does having better return alignment helps the final performance?
> >
> > i.e., can the problem of return alignment be solved by just having high target returns? (e.g. DT uses 5x maximum returns in the dataset)
>
> re:
>
> There is no guarantee that having better return alignment helps the final performance. We would like to emphasize that our primary goal is to align actual returns with desired target returns, as demonstrated in Figure 3. The improvement in final performance is merely a byproduct of this alignment and not a direct optimization target.
>
> We further visualize the effect of value alignment in Figure 5 (see appendix C.3, 777-790), showing 0x to 3x the maximum returns in the dataset. Doctor achieves much better alignment across a range of target returns compared to DT and MTM.
>
> Return alignment cannot be solved by just having high target returns. Doctor can achieve better alignment to some extent outside the dataset, but its alignment ability cannot extend too far beyond the maximum return seen during training.
> Figure 5 shows that the performance of Doctor saturates when the target return is set too high such as 3x maximum return.
>
>
>
> > Q2. Does having better return alignment helps online fine-tuning?
> >
> > It will be exciting if Doctor is able to show clear improvements in online fine-tuning experiments on AntMaze or Kitchen or Adroit tasks, where offline-to-online methods (e.g. Cal-QL) have showed significant improvements.
>
> re:
> Yes, better return alignment contributes to more effective online fine-tuning according to the results in table 4, where doctor achieved clear improvements on Adroit.
>
>
> > Q3. Other than Q1 (final performance) and Q2 (online fine-tuning), can you share your ideas on why having better return alignment is beneficial, if there is any?
>
> re:
>
> **Motivation**: In contrast to prior work, which focuses on improving final performance, our motivation is to extend this to a range of desired returns. Achieving this allows us to extract various levels of performance policies, which is particularly valuable for scenarios like game AI, where NPCs with diverse skill levels are essential for creating balanced and engaging gameplay. From this perspective, the improvement in final performance is merely a byproduct of this alignment.
>
> Regrading the effort of searching for the target return used in inference, the answer is yes. Previous works cannot generate a trajectory that achieves the desired return, as shown in our experiments in Figure 1 and 3, thus they need to search for the target return. In contrast, Doctor can generate a trajectory that achieves the desired return within the dataset and beyond to some extent, thus reducing the effort of searching for the target return.
>
> >Q4. Does Doctor improves stitching capability?
>
> re:
>
> Yes, as illustrated above.
>
> > Q5. What target return is used for evaluation?
>
> re:
>
> Yes, our target return is set based on the n times maximum return from the dataset as used by DT. Specifically, it is calculated as:
> target_return = min_return + (max_return-min_return)
>
> We hope these additional experiments and explanations address your concerns. Thank you for your valuable suggestions.

---

> > ### Comment · Reviewer_PDqo · 2024-11-22
> >
> > Thank you for the detailed response and additional experimental results.
> >
> > R1. Concern about Evaluation
> > * First, experiments in Appendix C.1 (especially Maze2D) clearly shows the stitching capability of Doctor.
> > * It **addresses the concern** on the evaluation of Doctor for its stitching capability.
> >
> > R2. Concern about Motivation
> > * For the response on the motivation, I agree that return alignment itself is an important problem.
> > * However, I still feel this point is not clear in the paper.
> > * **I strongly recommend emphasizing (clarifying) this motivation in the introduction section.**
> >
> > Based on the responses, I have increased the score to **6**.

---

> ### Author Response · Authors · 2024-11-22
>
> We appreciate your suggestion to further clarify the motivation. We will incorporate this in the next revision.
>
> Thank you again for your valuable feedback and for increasing the score.

---

### Official Review · Reviewer_p2GX · 2024-10-29

**Soundness:** 2
**Presentation:** 3
**Contribution:** 2
**Rating:** 3
**Confidence:** 4

**Summary:**

This paper introduces Double Checks the Transformer with Value Validation for Offline Reinforcement Learning (Doctor), a framework designed to address alignment issues between actual returns and desired target returns, especially in cases involving underrepresented or higher returns. Specifically, this approach integrates additional temporal-difference (TD) learning into the Transformer architecture, sampling actions around desired target returns and validating them through value functions. Experiments conducted in both offline and offline-to-online settings demonstrate improved alignment with target returns, showcasing the effectiveness of the proposed approach.

**Strengths:**

1. The paper is well-written and easy-to-read.
2. The motivation is clearly presented, focusing on addressing alignment issues within sequence modeling methods.

**Weaknesses:**

1. The paper lacks a comparison with state-of-the-art methods, such as RADT[1], which also addresses target return alignment, and QT[2], which enhances stitching capability by incorporating the TD method.
2. The performance improvements in D4RL are not significantly better than these baselines, diminishing the perceived effectiveness of the proposed framework. Although this work integrates the TD learning component into the MTM network, the improvement over MTM is marginal. Furthermore, during online fine-tuning, the performance gains are also less pronounced compared to the ODT baseline.

[1] Tsunehiko Tanaka, et al. Return-Aligned Decision Transformer. 2024

[2] Shengchao Hu, et al. Q-value Regularized Transformer for Offline Reinforcement Learning. 2024 ICML.

**Questions:**

1. Does interpolating between underrepresented returns and stitching information address the same issue? While stitching sub-optimal trajectories can yield better performance, it could also lead to undesirable outcomes not observed in the dataset, representing underrepresented returns. Thus, should the primary focus be on improving the stitching capability?
2. In line 167, $R_t$ is defined as the discounted return, while Decision Transformer (DT) treats it as the undiscounted sum of rewards. What is the rationale behind using the discounted return in this context?
3. In lines 185–186, how does the action-value $q_t$ provide the model with the ability to stitch sub-optimal trajectories?
4. Why was a bi-directional Transformer chosen, given that most DT-based approaches utilize a causal Transformer to predict actions in an auto-regressive manner? This causal structure is typically important during inference to predict future actions based on historical sequences.
5. Is the comparison of results in Table 1 equitable? The Doctor method requires sampling N target returns for alignment. Do the baseline methods also allow for such extensive sampling of target returns to construct their final results?
6. What is the time complexity of inference when compared to other baselines?

---

> ### Author Response · Authors · 2024-11-21
> **Part 1**
>
> **Weaknesses:**
>
> >1. The paper lacks a comparison with state-of-the-art methods.
>
> re:
>
> Thank you for pointing out. We have reproduced QT using the officially released [code](https://github.com/charleshsc/QT) and incorporated the results of RADT and QT into Table 3 (see appendix C.1 [703-729] in our revised version).
>
> >2. The performance improvements in D4RL are not significantly better than these baselines. Furthermore, during online fine->tuning, the performance gains are also less pronounced compared to the ODT baseline.
>
> re:
>
> Firstly, we would like to emphasize that our primary goal is to align actual returns with desired target returns, as demonstrated in Figure 3. The improvement in final performance is merely a byproduct of this alignment and not a direct optimization target. From this perspective, it is not surprising that Doctor does not significantly outperform other baselines that focus on policy improvement.
>
> To further showcase the effectiveness of our method, we have expanded our evaluation to include additional benchmark tasks, specifically Adroit and Maze2D. Notably, Doctor performs competitively across these tasks, demonstrating its robustness and generalizability.
>
> Regarding the online fine-tuning performance, since the initial scores are different for each method, the relative improvement may not be directly comparable. We compare the final performance, and Doctor achieves the best overall performance across tasks. Additionally, when the initial performance is similar, for example on Walker2D Medium in Table 2, Doctor achieves better performance during online fine-tuning than ODT.
>
>
> **Questions:**
>
> >1. Does interpolating between underrepresented returns and stitching information address the same issue? While stitching sub->optimal trajectories can yield better performance, it could also lead to undesirable outcomes not observed in the dataset, >representing underrepresented returns. Thus, should the primary focus be on improving the stitching capability?
>
> re:
>
> Interpolating between underrepresented returns and stitching are two different issues. Interpolating involves learning a robust policy that performs well when the data is sparse, while stitching focuses on combining segments to form better sequences. In our work, we aim to address both issues by leveraging the **double-check** mechanism, which allows the model to interpolate and extrapolate from the dataset.
>
> Regarding the undesirable outcomes due to stitching, this is precisely where the double-check mechanism comes into play. While supervised learning ensures that the generated actions remain within the distribution of the dataset, TD learning builds upon this foundation to improve extrapolation, thereby reducing the risk of generating unrealistic outcomes.
>
>
> >2. In line 167, is defined as the discounted return, while Decision Transformer (DT) treats it as the undiscounted sum of rewards. >What is the rationale behind using the discounted return in this context?
>
> re:
>
> We need to ensure consistency between the learning of the target return in the transformer and the action-value in the Q-function. We can use either discounted or undiscounted returns for both the transformer and the Q-function. We chose to use discounted returns.
>
> >3. In lines 185–186, how does the action-value provide the model with the ability to stitch sub-optimal trajectories ?
>
> re:
>
> In our method, the action-value function is trained via implicit Q-learning [1], an in-sample multi-step dynamic programming method that excels at stitching sub-optimal trajectories. We further enhance the stitching capability by incorporating our double-check mechanism, thereby reducing the risk of generating unrealistic outcomes.
>
> [1] Kostrikov, Ilya, Ashvin Nair, and Sergey Levine. "Offline reinforcement learning with implicit q-learning." arXiv preprint arXiv:2110.06169 (2021).

---

> ### Author Response · Authors · 2024-11-21
> **Part 2**
>
> **Questions**
>
> >4. Why was a bi-directional Transformer chosen, given that most DT-based approaches utilize a causal Transformer to predict >actions in an auto-regressive manner? This causal structure is typically important during inference to predict future actions based >on historical sequences.
>
> re:
>
> While early works like DT [1] and TT [2] use causal transformers, recent studies such as MTM [3] have shown that bidirectional transformers trained with a combination of random masking and autoregressive masking enable the model to learn a more generalized sequence representation and achieve better performance. Our experiments in Figure 3 also demonstrate that the BiT-based model MTM outperforms the CT-based model DT. Furthermore, our work based on BiT further improves the performance.
>
> [1] Chen, Lili, Kevin Lu, Aravind Rajeswaran, Kimin Lee, Aditya Grover, Misha Laskin, Pieter Abbeel, Aravind Srinivas, and Igor Mordatch. "Decision transformer: Reinforcement learning via sequence modeling." Advances in neural information processing systems 34 (2021): 15084-15097.
>
> [2] Janner, Michael, Qiyang Li, and Sergey Levine. "Offline reinforcement learning as one big sequence modeling problem." Advances in neural information processing systems 34 (2021): 1273-1286.
>
> [3] Wu, Philipp, Arjun Majumdar, Kevin Stone, Yixin Lin, Igor Mordatch, Pieter Abbeel, and Aravind Rajeswaran. "Masked trajectory models for prediction, representation, and control." In International Conference on Machine Learning, pp. 37607-37623. PMLR, 2023.
>
>
> >5. Is the comparison of results in Table 1 equitable? The Doctor method requires sampling N target returns for alignment. Do the >baseline methods also allow for such extensive sampling of target returns to construct their final results?
>
> re:
>
> Although Doctor samples N target returns for alignment, we process them as a batch and feed them into the model, resulting in relatively low computational overhead compared to other methods.
>
> Regarding the baseline methods, they are not suitable for the sampling strategy used by Doctor. This is because these methods typically lack a Q-function. Without a Q-function, even if target returns were sampled, these methods would have no mechanism to determine the optimal actions for the target return alignment.
>
> >6. What is the time complexity of inference when compared to other baselines?
>
> We have included a detailed comparison of inference speed and computational overhead with other models in the appendix. The results demonstrate that Doctor maintains efficient during inference compared to other approaches.
>
> | Time Complexity | DT    | MTM   | QT    | Doctor |
> |------------------|-------|-------|-------|--------|
> | Inference (seconds) | 0.01  | 0.056 | 0.016 | 0.065  |
> | Training (seconds)  | 2.13  | 1.27  | 2.51  | 1.34   |
>
>
> We hope these additions and clarifications address your concerns.

---

> > ### Comment · Reviewer_p2GX · 2024-11-27
> >
> > Thanks for the author's response.
> >
> > In terms of final task performance, the proposed method does not outperform the latest state-of-the-art approaches. While the author highlights the motivation of aligning with the target return, the necessity of this motivation is not adequately justified in the current version of the article. It is recommended that the next version provide further explanations or experimental evidence to substantiate the importance of this alignment. Additionally, there is a lack of comparison with other alignment methods, particularly in terms of alignment itself, rather than solely focusing on final performance. For these reasons, I have maintained the current score.

---

> > > ### Author Response · Authors · 2024-11-27
> > >
> > > Thanks for your detailed feedback.
> > >
> > > We have updated the latest version of the paper, where we have explained the necessity of the proposed value alignment method in the Introduction section. Additionally, we have added discussions of relevant papers into the Related Work section.
> > >
> > > We acknowledge that the current version lacks a direct comparison with other return alignment methods. However, we believe that the experiments presented in Figure 3 clearly demonstrate the superiority of Doctor in terms of return alignment. Specifically, Doctor ensures better interpolation between underrepresented returns in the dataset (as shown on the left side of the red line) and extrapolates to higher returns to some extent (as shown on the right side of the red line). This leads to superior alignment performance compared to previous RvS methods (closer adherence to the ideal line).
> > >
> > > We sincerely hope that our response can address your concerns.

---

### Official Review · Reviewer_LBGr · 2024-11-01

**Soundness:** 3
**Presentation:** 3
**Contribution:** 3
**Rating:** 5
**Confidence:** 1

**Summary:**

This paper proposes to combine supervised learning and temporal difference learning for offline reinforcement learning. First, the model is pretrained by randomly masking a subset of a trajectory and predicting it. In temporal difference learning, the model is trained to predict the action value. Experiments show promising results.

**Strengths:**

1. It looks reasonable to combine supervised learning and temporal difference learning for offline reinforcement learning.
2. The paper is easy to follow.
3. Experiments show promising results.

**Weaknesses:**

1. I am not sure about the novelty as self-supervised learning and value function learning are two common techniques.
2. Ablation studies about the design choices of the proposed framework are needed.
3. The performance over the baselines on different tasks seems inconsistent, as shown in Table 1.

**Questions:**

Please refer to the weakness.

---

> ### Author Response · Authors · 2024-11-21
> **Part 1**
>
> **Weaknesses:**
>
> >1. I am not sure about the novelty as self-supervised learning and value function learning are two common techniques.
>
> re:
>
> We acknowledge that self-supervised learning and value function learning are common techniques. Since all transformers are based on self-supervised learning and all value-based RL methods use value function learning, the novelty lies in the specific problem we are solving and how we solve it.
>
> **Problem:** Previous works focus solely on improving final performance. In contrast, our work aims to extend this to a range of desired returns, presenting a new and more challenging task. Achieving this allows us to extract various levels of performance policies, which is particularly valuable for scenarios like game AI, where NPCs with diverse skill levels are essential for creating balanced and engaging gameplay.
>
> **Method:** The core novelty of our work lies in the **double-check mechanism**, which combines self-supervised learning and value function learning. This enables the model to align predicted values with target returns more accurately while also enhancing its ability to fine-tune through further online exploration. Our method is the first to combine self-supervised learning and value function learning in this manner, and we believe this is a significant contribution to the field.
>
> We hope this clarifies the novelty of our work.
>
>
> >2. Ablation studies about the design choices of the proposed framework are needed.
>
> re:
>
> We explored the impact of different target return sampling sizes N in Figure 4, as N increases, Doctor achieves better
> alignment with the given target return.
> We also included the results for different transformer architectures in Figure 3, demonstrating that the BiT-based transformer (MTM) is a better choice than causal transformers (DT).
>
> >3. The performance over the baselines on different tasks seems inconsistent, as shown in Table 1.
>
> It is unrealistic to expect Doctor to outperform baselines on every individual task. This is consistent with observations in prior work, where different methods excel in different scenarios due to varying task characteristics. Furthermore, we would like to emphasize that our primary goal is to align actual returns with desired target returns. The improvement in final performance is merely a byproduct of this alignment.
>
>
> We hope these additions and clarifications address your concerns.

---

> > ### Comment · Reviewer_LBGr · 2024-11-26
> >
> > Thanks for the authors's response. I would like to keep my rating.

---

### Official Review · Reviewer_r5YX · 2024-11-08

**Soundness:** 3
**Presentation:** 3
**Contribution:** 2
**Rating:** 6
**Confidence:** 3

**Summary:**

The paper introduces a novel method called "Doctor" that integrates supervised learning (SL) and temporal difference (TD) learning to improve the alignment of predicted actions with desired target returns in offline reinforcement learning (RL). The method uses a bidirectional transformer and a double-check mechanism during inference to validate actions with value functions, ensuring better alignment and improved performance. The work is evaluated on the D4RL benchmark, demonstrating competitive performance compared to existing RvS-based and TD-based methods.

**Strengths:**

1. The core idea of the paper is interesting and well-motivated, addressing the challenge of aligning predicted actions with desired returns in offline RL.
2. The paper is well-written and easy to follow, making it accessible to readers with a background in RL and transformers.
3. The experimental results on the D4RL benchmark show that the proposed method, Doctor, outperforms several baselines in some tasks, indicating its potential effectiveness.

**Weaknesses:**

1. Model Choice (Bidirectional vs. Causal Transformer): The use of a bidirectional transformer, rather than a causal transformer, is not fully justified. Since inference is typically performed in a causal manner, it would be beneficial to provide more insight into why a bidirectional transformer was chosen and how it impacts the model's performance during inference.
2. Limitations Discussion: The paper lacks a detailed discussion of the limitations of the proposed work. It would be valuable to acknowledge potential drawbacks or scenarios where the method might not perform as well, setting realistic expectations for future research.
3. Inference Complexity: The inference complexity of the proposed method may be significantly higher than that of DT. A detailed comparison of the inference speed and computational overhead relative to other models would strengthen the paper.
4. Performance and Overhead: While the performance is competitive, the gains are marginal compared to the baselines. Given the higher computational cost, a more in-depth analysis of the trade-offs between performance and computational resources is necessary.
5. Evaluation Scope: The evaluation is conducted only on the D4RL benchmark, which may introduce a bias in the results. To ensure the robustness and generalizability of the proposed method, it would be valuable to conduct additional experiments on other offline RL benchmarks.
6. Reward Modeling Assumption: if I understand correctly, the underlying assumption is that value prediction is more general/robust than reward modeling for aligning the ground-truth reward during inference, which needs further theoretical justification. An analysis or explanation supporting this claim would add depth to the paper.

**Questions:**

1. Have you test your method in some robotics benchmarks?
2. Can the inference process be accelerated?

---

> ### Author Response · Authors · 2024-11-21
> **Part 1**
>
> **Weaknesses:**
> >1. Model Choice (Bidirectional vs. Causal Transformer): The use of a bidirectional transformer, rather than a causal transformer, is >not fully justified.
>
> re:
>
> While early works like DT [1] and TT [2] use causal transformers, recent studies such as MTM [3] have shown that bidirectional transformers trained with a combination of random masking and autoregressive masking enable the model to learn a more generalized sequence representation and achieve better performance. Our experiments in Figure 3 also demonstrate that the BiT-based model MTM outperforms the CT-based model DT. Furthermore, our work based on BiT further improves the performance.
>
> [1] Chen, Lili, Kevin Lu, Aravind Rajeswaran, Kimin Lee, Aditya Grover, Misha Laskin, Pieter Abbeel, Aravind Srinivas, and Igor Mordatch. "Decision transformer: Reinforcement learning via sequence modeling." Advances in neural information processing systems 34 (2021): 15084-15097.
>
> [2] Janner, Michael, Qiyang Li, and Sergey Levine. "Offline reinforcement learning as one big sequence modeling problem." Advances in neural information processing systems 34 (2021): 1273-1286.
>
> [3] Wu, Philipp, Arjun Majumdar, Kevin Stone, Yixin Lin, Igor Mordatch, Pieter Abbeel, and Aravind Rajeswaran. "Masked trajectory models for prediction, representation, and control." In International Conference on Machine Learning, pp. 37607-37623. PMLR, 2023.
>
> >2.Limitations Discussion: The paper lacks a detailed discussion of the limitations
>
> re:
>
> One potential limitation of our approach is that we additionally learn action-value heads, which introduces extra computational overhead during inference. We summarize the inference time and computational overhead (see appendix D.3 [848-856] in our revised version).
>
> | Time Complexity | DT    | MTM   | QT    | Doctor |
> |------------------|-------|-------|-------|--------|
> | Inference (seconds) | 0.01  | 0.056 | 0.016 | 0.065  |
> | Training (seconds)  | 2.13  | 1.27  | 2.51  | 1.34   |
>
>
> Another limitation is that since the goal of our work is to align actual returns with desired target returns, the improvement in final performance is only a byproduct of this alignment. The model may not always outperform other baselines that focus on policy improvement for every individual task, as shown in Table 1.
>
> >3.Inference Complexity: The inference complexity of the proposed method may be significantly higher than that of DT.
>
> re:
>
> During inference, Doctor processes one batch at a time and utilizes a unified architecture to generate both action predictions and value estimates simultaneously, enabling fully parallel computation.
> To address the concern, we have included a detailed comparison of inference speed and computational overhead with other models in the appendix (see appendix D.3 [848-856] in our revised version). The results demonstrate that Doctor maintains efficient during inference compared to other approaches.
>
> >4.Performance and Overhead: While the performance is competitive, the gains are marginal compared to the baselines.
>
> re:
>
> The computational overhead of our model is indeed higher compaired with MTM due to the inclusion of value function training. However, we have found this increase to be within an acceptable range.
>
> We have also conducted further experiments on some extra tasks (see appendix C.1 [703-729] in our revised version). For instance, in tasks like Maze2D, which emphasize the model's ability to effectively "stitch" together different parts of the sequence, Doctor outperforms some other baselines by a clear margin. Thus, we believe this additional cost is justified by the better alignment and on par or superior performance in a variety of tasks.
>
> >5.Evaluation Scope: The evaluation is conducted only on the D4RL benchmark, which may introduce a bias in the results.
>
> re:
>
> We have expanded our evaluation to include additional benchmark tasks, specifically Adroit, Maze2D. These tasks cover a broader range of environments, each emphasizing different aspects of the model’s ability. We found that Doctor's performance is competitive across these tasks, demonstrating its robustness and generalizability. We have included a detailed discussion of these results in the revised manuscript.
>
> >6.Reward Modeling Assumption
>
> re:
>
> We would like to clarify that our method does not rely on the 'Reward Modeling Assumption'. The better alignment and more robust performance of Doctor come from the double-check mechanism. Previous works such as DT assume that the return-conditioned learned model can generate a trajectory that achieves the desired return. We demonstrate that this is not the case, as previous works only care about improvement in the final performance and ignore the alignment assumption. In our work, we introduce a Q-function to double-check the alignment with the target returns, serving as a twofold verification (see appendix A in our revised version for more explanation.

---

> ### Author Response · Authors · 2024-11-21
> **Part2**
>
> **Questions:**
>
> 1.Have you test your method in some robotics benchmarks?
>
> re:
>
> Yes, Gym locomotion tasks are considered robotics benchmarks. To further validate our method, we have conducted additional experiments on maze2d and adroit tasks (see appendix C.1).
>
> 2.Can the inference process be accelerated?
>
> re:
>
> Since Doctor processes one batch at a time thus the computational overhead is relatively low as shown in Table. Please refer to the respond to the weakness.

---

> > ### Comment · Reviewer_r5YX · 2024-11-27
> >
> > Thanks for the response. I increase the score to 6. Good luck.

---

> > > ### Author Response · Authors · 2024-11-27
> > >
> > > Thank you for increasing the score. We appreciate your feedback.

---

### Author Response · Authors · 2024-11-26

**Global Response**

We sincerely thank all the reviewers for their valuable feedback on our work.

To address common questions and share additional experimental results, we have prepared a global response covering the following key aspects:

**1: Additional experimental results**

To further demonstrate the effectiveness of our method, we have expanded our evaluation to include additional benchmark tasks. Specifically, we have added three Maze2D tasks and three Adroit tasks in the offline setting as well as three Adroit tasks in the offline-to-online setting.

We have also compared our method with additional baselines, including the state-of-the-art method [1]. Furthermore, we provide additional visualizations of the alignment experiment [776-790], along with evaluations of inference time and training overhead [847-856].

All new results are included in the revised paper.

[1] Shengchao Hu, et al. Q-value Regularized Transformer for Offline Reinforcement Learning. 2024 ICML.

**2: Why Doctor acheives on par performance in some tasks?**

We would like to emphasize that our primary goal is to align actual returns with desired target returns, as demonstrated in Figure 3. The improvement in final performance is merely a byproduct of this alignment and not a direct optimization target. From this perspective, it is not surprising that Doctor does not significantly outperform other baselines that focus on policy improvement.

We would like to emphasize our motivation and novelty as follows:

**Motivation**: Previous works focus solely on improving final performance. In contrast, our work aims to extend this to a range of desired returns, presenting a new and more challenging task. Achieving this allows us to extract various levels of performance policies, which is particularly valuable for scenarios like game AI, where NPCs with diverse skill levels are essential for creating balanced and engaging gameplay.

**Novelty**: The core novelty of our work lies in the double-check mechanism, which combines self-supervised learning and value function learning. This enables the model to align predicted values with target returns more accurately while also enhancing its ability to fine-tune through further online exploration. Our method is the first to combine self-supervised learning and value function learning in this manner, and we believe this is a significant contribution to the field.

**3: Revised paper**


We have incorporated the reviewers' suggestions into the revised paper and uploaded it, with all major updates highlighted in blue for your convenience. The key revisions are as follows:

1) We have provided additional justification of value alignment (in appendix A).

2) Our experiments have been extended, comparing with additional baselines across more benchmark tasks (in appendix C).

3) We have included new visualizations for the alignment experiment, along with evaluations of inference time and training overhead (in appendix D.3).

4) More related papers have been cited.

Thank you again for your feedback. We are looking forward to hearing your thoughts.

---

### Meta-Review · Area_Chair_djAz · 2024-12-06

**Metareview:**

This work proposes a method called Double Checks the Transformer with value validation for offline RL. This method integrates supervised learning (SL) and temporal difference (TD) learning to improve the alignment of predicted actions with desired target returns in offline reinforcement learning (RL). The work is evaluated on the D4RL benchmark, demonstrating competitive performance compared to existing RvS-based and TD-based methods.

After the rebuttal and discussion, this paper receives review scores of 3, 5, 6, 6. Reviewers raised several concerns:
1) lack of comparisons with recent methods
2) The performance on D4RL is not significantly better than other methods.
3) The motivation for aligning with the target return is not adequately justified in the current version of the article.
4) a lack of ablation study
5) novelty is limited.

There are discussions between the reviewers and the authors. The reviewers who gave negative comments both replies and weren't fully satisfied with the responses from the authors. AC has checked the submission, the reviews, the rebuttal, and discussion. AC sided with the negative reviewers and believed this work needs further effort to revise and improve the method performance. Thus, a rejection is recommended.

**Additional Comments On Reviewer Discussion:**

There are discussions between the reviewers and the authors. Two reviewers are not fully convinced; the expert reviewer with a 3 score gave the following reasons to argue against accepting this paper. He or she eventually championed on the rejection.

In terms of final task performance, the proposed method does not outperform the latest state-of-the-art approaches. While the author highlights the motivation of aligning with the target return, the necessity of this motivation is not adequately justified in the current version of the article. It is recommended that the next version provide further explanations or experimental evidence to substantiate the importance of this alignment. Additionally, there is a lack of comparison with other alignment methods, particularly in terms of alignment itself, rather than solely focusing on final performance. For these reasons, I have maintained the current score.

---

### Decision · Program_Chairs · 2025-01-22

Reject